# Assessment of Two Commercial Comprehensive Gene Panels for Personalized Cancer Treatment

**DOI:** 10.3390/jpm13010042

**Published:** 2022-12-26

**Authors:** Anine Larsen Ottestad, Mo Huang, Elisabeth Fritzke Emdal, Robin Mjelle, Veronica Skarpeteig, Hong Yan Dai

**Affiliations:** 1Department of Clinical and Molecular Medicine, Faculty of Medicine and Health Sciences, Norwegian University of Science and Technology (NTNU), 7030 Trondheim, Norway; 2Department of Oncology, St. Olavs Hospital, Trondheim University Hospital, 7030 Trondheim, Norway; 3Department of Biosciences and Nutrition, Karolinska Institute, 17177 Stockholm, Sweden; 4Department of Pathology, Clinic of Laboratory Medicine, St. Olavs Hospital, Trondheim University Hospital, 7030 Trondheim, Norway

**Keywords:** next-generation sequencing, comprehensive cancer panel, personalized treatment

## Abstract

(1) Background: Analysis of tumor DNA by next-generation sequencing (NGS) plays various roles in the classification and management of cancer. This study aimed to assess the performance of two similar and large, comprehensive gene panels with a focus on clinically relevant variant detection and tumor mutation burden (TMB) assessment; (2) Methods: DNA from 19 diagnostic small cell lung cancer biopsies and an AcroMetrix™ assessment sample with >500 mutations were sequenced using Oncomine^™^ Comprehensive Assay Plus (OCAP) on the Ion Torrent platform and TruSight Oncology 500 Assay (TSO500) on the Illumina platform; (3) Results: OCAP and TSO500 achieved comparable NGS quality, such as mean read coverage and mean coverage uniformity. A total of 100% of the variants in the diagnostic samples and 80% of the variants in the AcroMetrix™ assessment sample were detected by both panels, and the panels reported highly similar variant allele frequency. A proportion of 14/19 (74%) samples were classified in the same TMB category; (4) Conclusions: Comparable results were obtained using OCAP and TSO500, suggesting that both panels could be applied to screen patients for enrolment in personalized cancer treatment trials.

## 1. Introduction

Analysis of tumor DNA and RNA by next generation sequencing (NGS) has been implemented in most diagnostic laboratories and plays various roles in cancer management. Detection of a specific variant can support pathologists in determining the correct differential diagnosis or enable clinicians to select the most effective treatment. Some tumor DNA variants lead to treatment with targeted drugs that interacts with the mutated protein’s functional site. These are typically activating mutations in oncogenes, such as *EGFR* and *BRAF* [1,2]. Other tumor DNA mutations cause protein changes that cannot be targeted directly, but the mutated protein acts in a cellular pathway that can be targeted. For example, mutations in the DNA homologous recombinational repair (HRR) pathway can be targeted with a drug that inhibits the DNA damage sensor function of the PARP protein, regardless of which protein is mutated [3]. Targeted NGS can also be applied to evaluate tumor mutation burden (TMB), i.e., the number of mutations per megabase of the genome. Higher TMB leads to higher neoantigen production and has been associated with response to immunotherapy in some studies [4]. 

Analysis of tumor DNA by NGS may also expand the patient’s treatment options by identifying molecular characteristics that qualify for enrollment in a clinical trial. Clinical trials of new targeted drugs are commonly designed as basket or umbrella trials where patients are enrolled based on a wide selection of mutations [5,6,7]. For example, an ongoing nation-wide Norwegian basket trial on precision medicine enrolls previously treated advanced cancer patients based on detection of a tumor mutation that can be targeted by the drugs available through the trial (currently 13 drugs, IMPRESS-Norway [8]). Additionally, patients with high TMB may receive immunotherapy through this study. 

Screening for such clinical trials may require analysis of tumor DNA by NGS using a large gene panel. The TruSight Oncology 500 Assay (TSO500) from Illumina is currently applied in screening for IMPRESS-Norway. To date, few diagnostic laboratories in Norway apply large gene panels for routine diagnostics, since small, focused gene panels (<50 genes) are sufficient to identify targets for drugs that are reimbursed at our country’s public hospitals. Furthermore, many diagnostic laboratories have established NGS on the Ion Torrent platform and cannot apply the Illumina platform-specific panel TSO500. Few laboratories have access to both platforms.

This study aimed to compare the performance of two large, commercially available and platform-specific NGS panels: Oncomine^™^ Comprehensive Assay Plus (OCAP) for Ion Torrent and the TSO500 panel for Illumina. OCAP targets regions in 497 genes for DNA analysis and 49 genes for RNA analysis. The clinical utility of this panel has previously been assessed [9]. TSO500 targets 523 genes for DNA analysis and 55 genes for RNA analysis. Both OCAP and TSO500 enable assessment of TMB.

NGS libraries were prepared from 19 diagnostic formalin-fixed paraffin-embedded (FFPE) samples from small cell lung cancer (SCLC) and an AcroMetrix™ assessment sample with >500 mutations with variant frequencies (VAFs) in the range of 5–35%. The NGS quality was compared between OCAP and TSO500 by assessing the mean coverage and coverage uniformity across the target region, followed by a closer comparison of the mean coverage in 35 targetable oncogenes, 26 genes in the HRR pathway, and the ten most frequently mutated genes in SCLC. The sensitivity and specificity were further evaluated by comparing detected pathogenic variants and their VAFs and the TMB scores. RNA analysis was briefly assessed by sequencing a constructed sample with 15 different fusions.

## 2. Materials and Methods

### 2.1. DNA and RNA Material

Nineteen diagnostic FFPE tumors from SCLC patients (17 limited disease stage and two extensive disease stage) were retrieved from a regional biobank, Biobank 1. The Norwegian Regional Committee for Medical and Health Research Ethics (REC) Central, the Norwegian Health Department, and the Norwegian Data Protection Authority approved the biobank. The results of this study did not influence the treatment of these patients.

DNA and RNA were extracted together using Allprep DNA/RNA FFPE kit (Cat. no: 80284, Qiagen, Hilden, Germany) or separately using QIAamp DNA FFPE Tissue Kit (Cat. no: 56404, Qiagen, Hilden, Germany) for DNA extraction and RNeasy FFPE kit (Cat. no: 73504, Qiagen, Hilden, Germany) for RNA extraction. Nucleic acid concentration was measured fluorometrically by Qubit^®^ Fluorometric Quantification (Cat. no: Q33265 and Q10210, Thermo Fisher Scientific, Waltham, MA, USA). RNA quality assessment was performed by quantitative real-time PCR of a 258 bp fragment in the gene *ACTB*. Quality assessment of DNA was performed by quantitative real-time PCR of a 300 bp fragment in the gene *FCGR3B*, as described previously [10]. 

Thermo Scientific™ AcroMetrix™ Oncology Hotspot Control (Cat. no: 969056, Thermo Fisher Scientific, Waltham, MA, USA), was used as a quality assessment sample for DNA sequencing. This artificially constructed DNA sample contained 521 somatic mutations with theoretical VAFs of 5–35% in 53 genes, including clinically relevant mutations. Seraseq^®^ FFPE NTRK Fusion RNA (Cat. no: 0710-1031, SeraCare Life Sciences, Milford, CT, USA) was used as a quality assessment sample for RNA sequencing and contained 15 different fusions involving the *NTRK* gene. 

### 2.2. NGS Library Preparation and Sequencing

NGS libraries using both OCAP and TSO500 were prepared from the SCLC samples and the assessment samples. Following the protocol for Oncomine^™^ Comprehensive Assay Plus (OCAP, cat. no: A49667 and A49671, Thermo Fisher Scientific, Waltham, MA, USA), 20–100 ng SCLC DNA and 2 ng and 20 ng of the AcroMetrix™ assessment sample were treated with Uracil DNA Glycosylase (Cat. no: 78310100UN, Thermo Fisher Scientific, Waltham, MA, USA) to remove deaminated cytosines that would cause a non-biologically relevant C > T transition. cDNA was synthesized from 100 ng SCLC RNA and 20 ng Seraseq^®^ assessment RNA using Ion Torrent^™^ NGS Reverse Transcription kit (Cat. no: A45003, Thermo Fisher Scientific, Waltham, MA, USA). Then, eight DNA and eight cDNA samples were subjected to each NGS library preparation run using the Ion Chef™ System for automatic preparation that was already implemented in routine use at the pathology department at St. Olavs Hospital. 

In brief, the target region was amplified by PCR using gene-specific primer pairs, with separate primer pools for the DNA and RNA libraries. The barcoded adapters were ligated to the PCR fragments, and the barcoded libraries were then purified, normalized by beads, and pooled. The library pool was quantified using the Ion Library TaqMan^™^ Quantification Kit (Cat. no: 4468802, Thermo Fisher Scientific, Waltham, MA, USA) and subsequently diluted to the appropriate concentration for the templating reaction, in which library fragments were attached to the Ion Sphere Particles (ISPs) and amplified by emulsion PCR. Template-positive ISPs were then loaded onto Ion 550^™^ chips and sequenced using Ion GeneStudio^™^ 5S Prime System. Data was automatically transferred, and reads mapped to the human genome assembly 19 by Torrent Suite v.5.16 (Thermo Fisher Scientific, Waltham, MA, USA). Variants were annotated using the Ion Reporter™ Software v.5.18.2.1 (Thermo Fisher Scientific, Waltham, MA, USA) with the databases NCBI dbSNP (v.154) and ClinVar v.20201121 for annotation. 

Following the protocol for Illumina TruSight 500 Oncology Assay (TSO500, cat. no: 20028215, Illumina, San Diego, CA, USA), eight DNA and eight cDNA samples were subjected to each manual NGS library preparation run. A mass of 65–100 ng SCLC DNA and 40 ng of the AcroMetrix™ assessment sample were fragmented to 90–250 bp using Covaris M220 Focused-ultrasonicator (Covaris LLC, Woburn, MA, USA). cDNA was synthesized from 100 ng SCLC RNA and 40 ng Seraseq^®^ assessment RNA following the library preparation protocol. 

Further, the DNA and cDNA were end-repaired and A-tailed before sequencing barcodes, including a unique molecular index (UMI), were ligated to each fragment. The barcoded DNA was then amplified by PCR. The target region was selected by adding target-specific biotin-bound probes that hybridized to library fragments. Separate pools of probes were used for the preparation of DNA and RNA libraries. The probe-library fragments were captured by streptavidin beads, and the hybridization capture procedure was repeated. The captured libraries were then amplified by PCR and individually quantified using Collibri™ Library Quantification Kit (Cat. no: A38524500, Thermo Fisher Scientific, Waltham, MA, USA). Bioanalyzer High Sensitivity DNA Analysis (Cat. no: 5067-4626, Agilent Technologies, Santa Clara, CA, USA) was used to assess the library quality. Finally, libraries were normalized by dilution and pooled into one DNA library pool and one RNA library pool, which were subsequently pooled 4:1, loaded onto NS500 High Output flowcell (Cat. no: 20024908, Illumina, San Diego, CA, USA), and sequenced using NextSeq 550 (Illumina, San Diego, CA, USA). Fastq files were loaded into CLC Genomic Workbench (v22.0.1, Qiagen, Hilden, Germany). This software is panel agnostic, has a visual user interface and requires less bioinformatic expertise than the panel-specific software Illumina Local App. Fastq files were analyzed using the ready-to-use TSO500 workflow. By default, reads were mapped to the human genome assembly 38 and annotated with NCBI dbSNP (v.151) and ClinVar (v. 20210821). 

### 2.3. Assessment of NGS Quality Metrics

NGS quality was evaluated by assessing the mean number of reads mapped to the reference genome per sample, mean percentage of reads mapped in the target region, mean base coverage, and mean coverage uniformity across the target region. To evaluate the read coverage more closely, we selected three sets of genes where the coverage was assessed in the overlapping target region. The first gene set comprised targetable oncogenes, and we selected the 35 genes targeted by the Oncomine^™^ Focus Assay (Thermo Fisher Scientific). This panel is considered a large targetable panel for solid cancers compared to TruSight Tumor 15 (Illumina, San Diego, CA, USA), which targets 15 genes and QIAseq Targeted DNA Human Actionable Solid Tumor Panel (Qiagen, Hilden, Germany), which targets 22 genes. The second gene set comprised 26 genes covered by the Oncomine^™^ HRR Pathway Predesigned Panel (Thermo Fisher Scientific, Waltham, MA, USA). Since the diagnostic DNA samples were from SCLC patients, the third gene set comprised the ten most frequently mutated genes in SCLC, according to My Cancer Genome [11]. Python (v.3.7) was used to calculate the mean coverage for each gene and prepare the figures. 

### 2.4. Sensitivity and Specificity of Variant Detection

Variant detection was restricted to the overlapping target region. Using Python (v. 3.7) for OCAP and Rstudio (v. 2022.02.3) for TSO500, variants were annotated according to a set of criteria outlined below. Each variant was then classified as “pass” if all criteria were met or “fail” if at least one criterion was not met. For simplicity, we included all variants classified as ‘pathogenic/likely pathogenic’ by either OCAP, TSO500, or both when the panels were compared, since they applied different versions of ClinVar for annotation. The genomic positions of variants detected by TSO500 were converted to hg19 positions using UCSC LiftOver. 

The following criteria were applied for both OCAP and TSO500:Variant type = SNV, MNV or indelLocation = ‘splice site’ or ‘exonic’Nucleotide length ≥1Variant effect = ‘non-synonymous’Allele frequency ≠ 100Homopolymer length < 5Phred score ≥ 200Common SNP = ‘No’ClinVar category = ‘pathogenic’ or ‘likely pathogenic’

Three additional criteria were applied for OCAP only: Coverage > 100Filter = ‘Pass’*p*-value < 0.01

One additional criterion was applied for TSO500 only:Singleton UMI or Big UMI > 0

### 2.5. TMB

TMB for OCAP was evaluated by Ion Reporter and TMB for TSO500 was evaluated by CLC Genomic Workbench and Illumina Local App using the default settings. The cut-offs for high TMB (>20 mut/Mb), intermediate (5–20 mut/Mb) and low TMB (<5 mut/Mb) were applied according to the protocol for IMPRESS-Norway [8]. 

## 3. Results

### 3.1. DNA Target Region of OCAP and TSO500

OCAP targeted 815 kb in 497 genes and was slightly smaller than TSO500, which targeted 944 kb in 523 genes. In total, 666 kb in 306 genes were targeted by both panels (Figure 1 and Appendix A). 

### 3.2. DNA NGS Library Preparation and Quality Metrics

OCAP libraries were prepared from median 40 ng DNA in two days with 3–4 h of hands-on time. TSO500 libraries were prepared from median 100 ng DNA in three days with 11 h of hands-on time. Table 1 shows NGS quality metrics for the DNA libraries in each run. TSO500 generated a higher mean number of reads mapping to the reference genome than OCAP (mean 33.2 mill vs. 26.5 mill). However, only 50% of the TSO500 reads mapped to the panel target region, compared to >90% of the OCAP reads. The remaining TSO500 reads mapped mainly to intronic regions between the exon target regions. The two panels achieved comparable mean base coverage in their respective target regions (2000× for OCAP and 1777× for TSO500). Additionally, both panels displayed high coverage uniformity (~95%), defined as the percentage of bases in the target region with >20% of the sample read coverage. The NGS quality was stable between runs for both OCAP and TSO500. 

### 3.3. Mean Read Coverage of Selected Clinically Relevant Genes

The mean read coverage was assessed in three sets of 64 total genes and restricted to the overlapping target regions (Figure 2, Figure 3 and Figure 4). The mean coverage in these genes (1941× for OCAP and 1261× for TSO500) was similar to the overall mean read coverage. Most genes (92% for OCAP, 73% for TSO500) achieved >1000× coverage, sufficient for confidently calling a variant at >5% VAF. Regardless of the panel, the coverage was more even in the AcroMetrix™ assessment sample than in the SCLC samples. The mean coverage in the SCLC samples represents 19 diagnostic samples with varying DNA quality.

The mean coverage of the gene *TERT* was noticeably low (24.5×) in the TSO500 libraries in both the SCLC samples and the AcroMetrix™ assessment sample (Figure 3). Reads were mapped to the promoter region, but no reads were mapped to the downstream region, even though it was included in the panel target region. Conversely, the OCAP library displayed high coverage in the downstream region of *TERT*, but almost no reads were mapped to the promoter region.

### 3.4. Sensitivity and Specificity in Variant Detection

Variant detection was restricted to the overlapping target region. Variants were annotated as “pass” or “fail” according to predefined criteria. Only variants that were classified as pathogenic/likely pathogenic by ClinVar were included since the aim was to evaluate the clinical utility of the panels. 

We counted the number of variants that were annotated as “pass” by both panels (=“complete match”), and “pass” by one panel and “fail” in the other (=“partial match”), and “pass” by one panel and not detected at all by the other (=“unmatched”). 

Twenty unique variants were detected in the SCLC DNA samples by OCAP and TSO500. All variants were “complete match”, and the two panels reported similar VAFs (adjusted R^2^ = 0.96, *p* < 0.001, Figure 5). 

The mutations were detected in *TP53* (*n* = 10), *RB1* (*n* = 3), *NFE2L2* (*n* = 2), *NF1* (*n* = 2), *NOTCH1* (*n* = 1), *PIK3CA* (*n* = 1) and *CREBBP* (*n* = 1). A proportion of 6/19 (32%) samples had one mutation, 4/19 (21%) had two mutations, 2/19 (11%) had three mutations, and 7/19 (37%) samples had no detectable mutations.

In the two AcroMetrix™ libraries prepared by OCAP using 20 ng and 2 ng DNA input, there were 137 “complete match” variants and 18 “partial match” variants (Figure 6). Most “partial match” variants (14/18) were “fail” in the run with 2 ng input. The median VAF of the “partial match” variants was 7.0%, and 12/18 (67%) variants were C > T or G > A transitions. All “partial match” variants were manually evaluated as likely true mutations. No variants were “unmatched” between the two OCAP runs. 

In the two AcroMetrix™ libraries prepared by TSO500 using 40 ng DNA input, there were 183 “complete match” variants and five “partial match” variants (Figure 6). The “partial match” variants had median VAF 5.1% and all were manually evaluated as likely true mutations. No variants were “unmatched” between the two TSO500 runs.

The variants detected in the AcroMetrix™ assessment sample were then compared between OCAP and TSO500. Of the total 195 unique variants, 157 variants (80%) were a “complete match”, and 13 variants (7%) were a “partial match” (Figure 6). A proportion of 11/13 “partial matched” variants were “pass” by TSO500 only (median VAF 8.6%), and 2/13 variants were “pass” by OCAP only (median VAF 3.4%). There was a significant positive correlation between the VAFs in OCAP and TSO500 (adjusted R^2^ = 0.68, *p* < 0.001, Figure 7), and the VAFs were slightly higher in OCAP than TSO500.

There were 25 “unmatched” variants that were annotated as “pass” by TSO500 only (median VAF 7.0%) and were not called by OCAP, although reads were observed at all positions. A proportion of 14/25 variants were manually observed in the OCAP reads, but the VAF was too low (<2%) to determine whether these were true variants. A proportion of 12/25 (48%) unmatched variants were C > T or G > A transitions. No known actionable variants were among the unmatched variants. We observed that 14/25 unmatched variants were located within 25 bp of another variant in the same read. Similarly, the remaining 11/25 variants were located close to other non-pathogenic variants given in the list from the manufacturer.

### 3.5. Evaluation of Tumor Mutation Burden (TMB)

The SCLC samples were classified as TMB-low (<5 mut/Mb), TMB-intermediate (5–20 mut/Mb) or TMB-high (>20 mut/Mb). The same TMB classification was reported by OCAP and TSO500 in 14/19 (74%) SCLC samples (Figure 8). The remaining 5/19 samples were defined as TMB-intermediate by OCAP and TMB-low by TSO500. Importantly, both panels identified the same SCLC sample (from patient 4) as TMB high. 

### 3.6. Fusion Detection

RNA sequencing for fusion detection was briefly assessed in this study. OCAP targeted 49 genes and TSO500 targeted 55 genes. A total of 28 genes were targeted by both panels (Figure 9 and Appendix A). A proportion of 14/15 NTRK fusions in the Seraseq^®^ FFPE NTRK Fusion sample were called by OCAP. The last fusion was detected, but was filtered out due to low read count (count 41). All 15 fusions were called by TSO500.

## 4. Discussion

Most diagnostic laboratories perform tumor DNA analyses by NGS using either the Ion Torrent or the Illumina platform. While gene panels are usually specific to one platform, similar panels may be available to the other platform. In this study, we assessed the performance of two large, platform-specific NGS gene panels that enable the detection of DNA variants, gene fusions, copy number variations, microsatellite instability, and TMB. Oncomine^™^ Comprehensive Assay Plus (OCAP) from Ion Torrent and TruSight Oncology 500 Assay (TSO500) from Illumina were assessed with a focus on the detection of clinically relevant variants in DNA and the evaluation of TMB. The results show high concordance between OCAP and TSO500 in detecting pathogenic and likely pathogenic variants. In 19 diagnostic FFPE samples from SCLC tumors, 100% of mutations were detected by both panels. Similarly, 80% of the detected mutations in an AcroMetrix™ assessment sample with >500 mutations and lower VAFs were detected by both panels. Both panels reported the same TMB classification in most SCLC samples (74%).

The two panels achieved comparable NGS quality regarding the mean base coverage and the coverage uniformity across the target regions. The NGS quality was similar to that of a previous study that applied OCAP for clinically relevant DNA sequencing [9]. Furthermore, both OCAP and TSO500 enabled NGS library preparation and sequencing of eight samples per run. The panels displayed stable overall quality between runs and stable variant detection in two replicate libraries prepared from the AcroMetrix™ sample. 

Notably, the reported VAFs were similar between the panels, both in the AcroMetrix™ sample and the SCLC samples. FFPE samples are considered more challenging starting material due to the varying DNA quality [12]. Similar VAF was observed even though UMIs were only incorporated in the TSO500 libraries. In line with this, we have previously experienced that UMIs are unnecessary for precise VAF reporting when the tumor cell content is above 5% (unpublished data). We observe that the VAF was somewhat higher by TSO500 than OCAP, but more importantly, the VAFs were in the same range.

Variants with low VAFs are commonly the most challenging ones to detect, which was the case in this study. A total of 20% of the detected variants in the AcroMetrix™ assessment sample were only detected by one panel, and the mean VAF of these variants was 6.3%. Most variants were only detected by TSO500, possibly because the input DNA amount used in the preparation of these libraries compared was twice as high in comparison to OCAP. We cannot exclude that some of the C > T or G > A transitions only detected by TSO500 may have been artifacts, since the DNA input in TSO500 library preparation was not treated with uracil DNA glycosylase. This enzyme may be used to remove deaminated cytosine that would lead to false C > T transition [13]. Additionally, we observed that many variants were located within 25 bp of another variant on the same read, i.e., on the same DNA fragment. The high number of mutations in this artificial sample increases the risk of allelic dropout in OCAP library preparation, since it involved target region amplification by PCR. 

Others have demonstrated high concordance between TMB scoring by TSO500 and whole exome sequencing [14,15] and the FDA-approved FoundationOneCDx [14]. In the diagnostic setting, TMB category, specifically the TMB-high category, determines whether the patient should be offered immunotherapy. Moreover, the treatment selection for patients with a TMB score close to the cut-off value may be evaluated in more detail based on other patient characteristics than TMB alone. The comparison between OCAP and TSO500 shows high concordance in TMB categorization, even though the panels utilize different genomic regions for evaluation. Notably, those samples with a discrepancy in the TMB category were TMB-low or in the lower half of TMB-intermediate, neither of which are likely to impact treatment selection. Furthermore, we show that the TMB category remained the same regardless of whether the TSO500 samples were analyzed by Illumina Local App or CLC Genomics Workbench. We note that a higher TMB score was reported by Illumina Local App, possibly because synonymous variants were counted by default, while these were excluded by CLC Genomics Workbench and Ion Reporter. 

There were some differences between the two panels as well. Preparing OCAP libraries required less input DNA and far less hands-on time than preparing TSO500 libraries, especially since we applied the automatic library preparation system for OCAP. While OCAP utilizes target region amplification by PCR, TSO500 utilizes hybridization capture. This principle is not as vulnerable for allelic dropout, but it is generally considered less efficient in target selection, and thereby less sensitive for detecting low frequency variants. Therefore, more input DNA was used in this protocol. Another difference was that UMIs were incorporated only in preparing TSO500 libraries, as previously mentioned. These are advantageous in evaluating low VAF variants because they enable quantification of the number of unique genomes sequenced [16]. We previously showed that assessing the tumor cell content and input DNA quality can serve as an alternative to UMI incorporation [10]. Lastly, the two sequencing technologies each have inherent limitations, such as Ion Torrent’s struggle with library preparation of GC-rich regions and calling indels in homopolymers, and Illumina enables shorter read length than Ion Torrent [17].

Only SCLC tumors were included in this study. These are typically high in tumor cell content compared to other tumor types. However, they still represent a challenging group, since the biopsies are usually small in size and vulnerable to over fixation in formalin, causing low DNA quality [13]. We assume that comparable results would be obtained with other tumor types, since the tumor cell content, input DNA quality, and quantity are likely more determinant than the type of cancer. On the other hand, it would be relevant to compare NGS of tumors known for other genomic aberrations beyond point mutations, such as chromosomal rearrangement, copy number variations, or gene fusions. To compensate for using only one tumor type, we included the AcroMetrix™ assessment sample, which represents a broader spectrum of mutations and simulates a sample with lower tumor cell content. 

There were other limitations as well. Reads from OCAP and TSO500 were analyzed with different software, causing variability between the panels. However, we believe that our results show flexibility in the choice of analytic software. We utilized CLC Genomics Workbench from Qiagen for the analysis of TSO500, which require less bioinformatic expertise than the software “Local App” provided by Illumina. Similarly, the Ion Torrent platform offers a streamlined workflow from library preparation to variant interpretation, which does not require bioinformatic expertise. Further, TMB evaluation was assessed without any adjustments in the default algorithms, indicating that TMB assessment is robust and seems independent of the selected genes. 

It was not expected, nor was it the goal, that the results should be identical between OCAP and TSO500, since the panels differ in design, library preparation chemistry, sequencing technology, and analysis software. We believe this variability is a strength of this study, since the results suggest that the tumor DNA analysis is flexible and can be tailored to the individual laboratory. Such flexibility is an advantage for clinical trials dependent on NGS screening being performed at several regional diagnostic laboratories.

In conclusion, this study shows that OCAP and TSO500 displayed comparable performance in clinically relevant DNA sequencing, suggesting that both panels could be applied to screen patients for enrolment in personalized cancer treatment trials. 

## Figures and Tables

**Figure 1 jpm-13-00042-f001:**
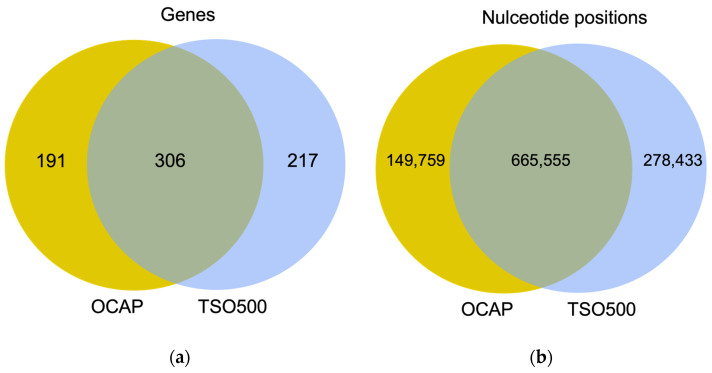
The number of (**a**) genes and (**b**) nucleotides that were targeted by OCAP (yellow), TSO500 (blue), and by both panels (green). A complete gene list is available in Appendix A.

**Figure 2 jpm-13-00042-f002:**
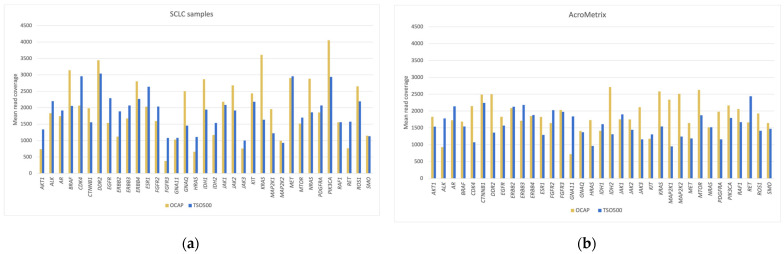
Mean read coverage in 35 targetable oncogenes that are included in the panel Oncomine^™^ Focus Assay, which is a relatively large panel of targetable genes. Mean read coverage is shown for (**a**) the SCLC samples and (**b**) the AcroMetrix™ assessment sample.

**Figure 3 jpm-13-00042-f003:**
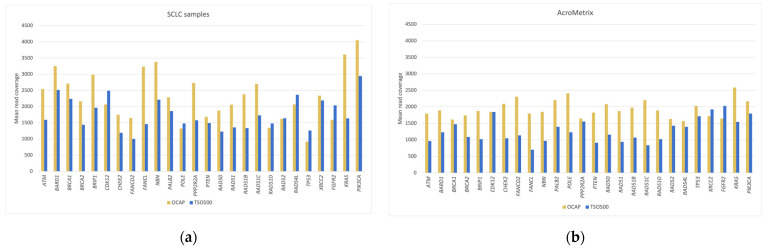
Mean read coverage in 26 genes in the DNA homologous recombinational repair (HRR) pathway. Mean read coverage is shown for (**a**) the SCLC samples and (**b**) the AcroMetrix™ assessment sample.

**Figure 4 jpm-13-00042-f004:**
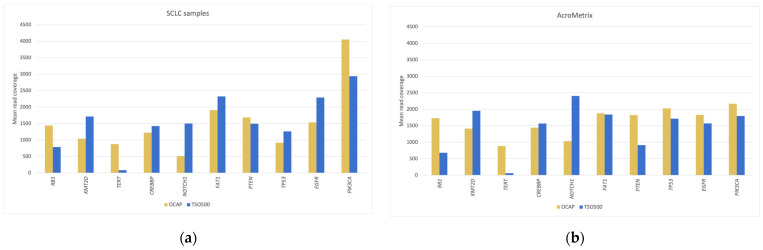
Mean read coverage in the ten most frequently mutated genes in SCLC. Mean read coverage is shown for (**a**) the SCLC samples and (**b**) the AcroMetrix™ assessment sample.

**Figure 5 jpm-13-00042-f005:**
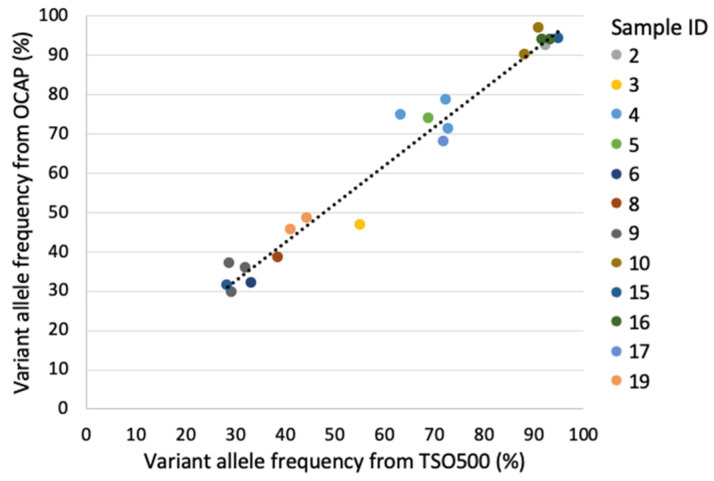
Variant allele frequency (VAF) of variants detected in the SCLC samples by OCAP and TSO500. Adjusted R^2^ = 0.96, *p* < 0.001.

**Figure 6 jpm-13-00042-f006:**
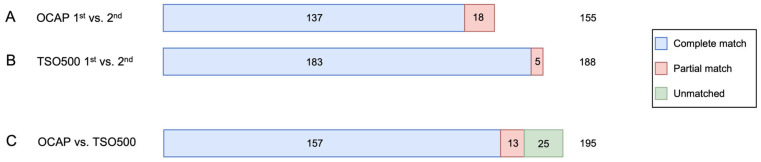
A: Number of variants detected in two OCAP libraries prepared from the AcroMetrix™ assessment sample. B: Number of unique variants detected in two TSO500 libraries prepared from the AcroMetrix™ assessment sample. C: Variants detected in the AcroMetrix™ assessment sample by OCAP, TSO500, and both panels. Variants were included in comparison if they were annotated as “pathogenic” or “likely pathogenic” by OCAP or TSO500 since different versions ClinVar were applied.

**Figure 7 jpm-13-00042-f007:**
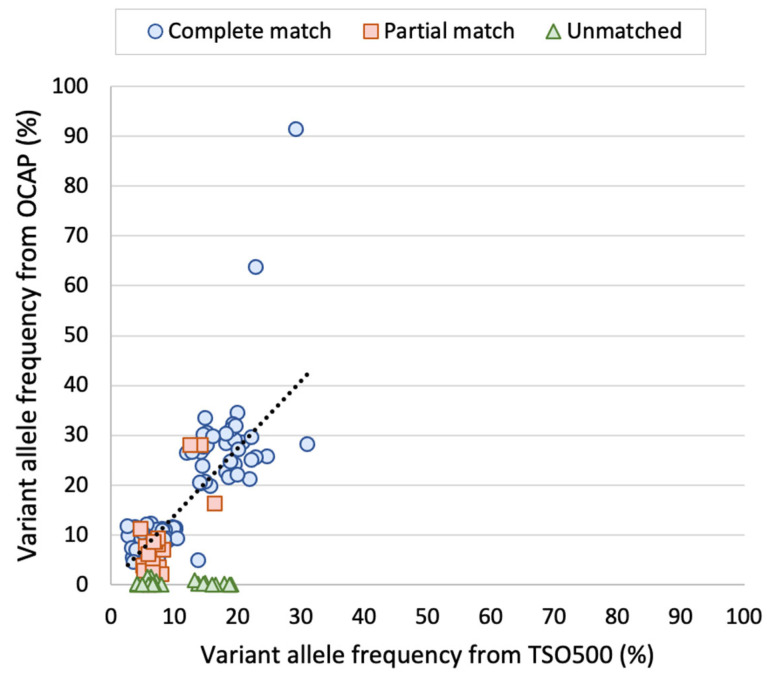
Variant allele frequency (VAF) of variants detected in the AcroMetrix™ sample by OCAP and TSO500. Two outliers and unmatched variants were excluded from the calculation of the regression line. Adjusted R^2^ = 0.68, *p* < 0.001.

**Figure 8 jpm-13-00042-f008:**
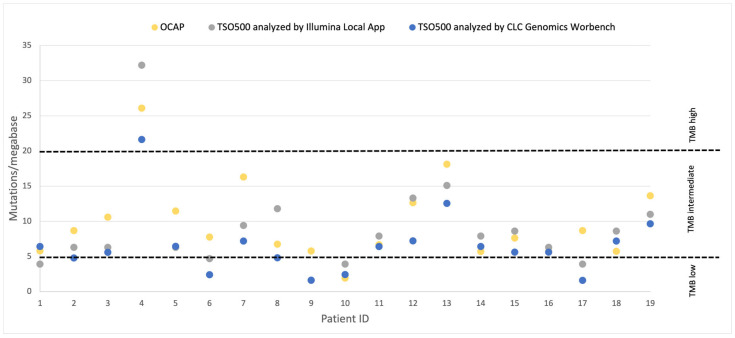
TMB score reported by OCAP and TSO500 in the SCLC samples. TMB was evaluated by two software: CLC Genomics Workbench (Qiagen) and Illumina Local App (Illumina).

**Figure 9 jpm-13-00042-f009:**
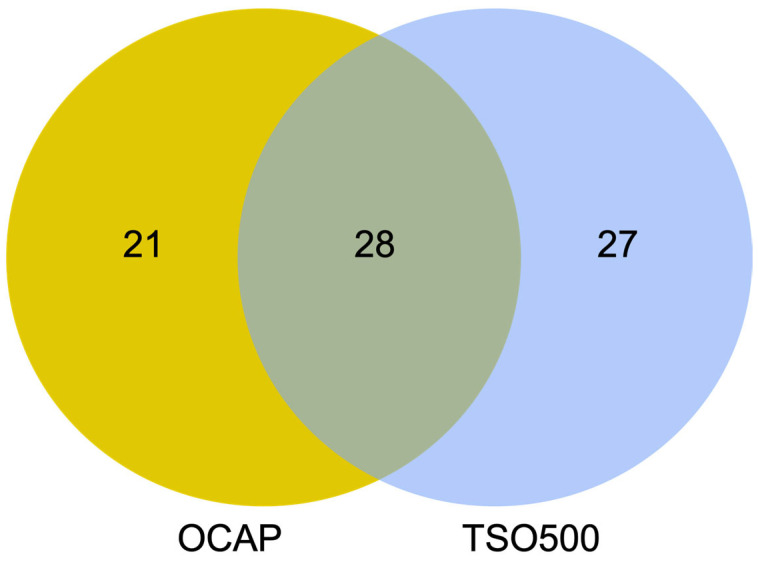
The number of genes included for gene fusion detection by OCAP (yellow), TSO500 (blue), and by both (green). The complete gene list is available in Appendix A.

**Table 1 jpm-13-00042-t001:** NGS quality metrics for each sequencing run.

Panel	Run Number	Mapped Reads (Mill.) ^1^	On Target (%) ^2^	Mean Base Coverage	Uniformity ^3^
OCAP	1	24.1 (±1.4)	93.4 (±0.9)	1841 (±112)	94.5 (±1.8)
	2	26.9 (±7.0)	90.7 (±0.8)	2095 (±542)	95.5 (±1.2)
	3	27.6 (±2.2)	91.3 (±1.6)	2143 (±196)	95.6 (±3.1)
	4	23.5 (±3.3)	92.2 (±0.8)	1811 (±256)	94.7 (±4.6)
TSO500	1	35.3 (±7.8)	52.8 (±4.2)	1273 (±127)	95.8 (±0.5)
	2	28.2 (±6.2)	50.5 (±2.3)	2218 (±542)	91.8 (±2.7)
	3	37.6 (±8.6)	51.7 (±5.3)	1981 (±127)	91.9 (±2.3)

Numbers represent the mean value of libraries included in the study from each run ± standard deviation. ^1^ Paired reads from TSO500 are counted as one. ^2^ On Target: The percentage of reads that mapped panel’s target region. ^3^ Uniformity: The percentage of positions with at least 20% of the mean coverage.

## Data Availability

Not applicable.

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
