# Peer review of "Assessment of Two Commercial Comprehensive Gene Panels for Personalized Cancer Treatment"

_jpm, 2022, doi:10.3390/jpm13010042_

Round 1

Reviewer 1 Report

The article 'Assessment of two commercial comprehensive gene panels for personalized cancer treatment' by Anine Larsen Ottestad et al, describes the assessment of two similar and large gene panels with a focus on variant detection and tumor mutation burden analysis. OCAP and TSO500 methods were used for detection and found that both methods reported similar variant allele frequency, suggesting that both panels could be used for screening patients during enrollment of personalized cancer treatment trials. The article has been well written and only a few minor details given below must be added to improve the quality of the work:

1. Provide catalog numbers of the kits/reagents used, with the company names wherever possible.

2. SCLC tumors have been studied, what are the authors' thoughts on using other tumor types, and what could be the potential outcome of the results? Please describe briefly in the Discussion section.

Author Response

We would like to thank the reviewer and express our appreciation for the feedback. In accordance with the response, we have included catalogue numbers and company names in the Methods section. To increase the readability, we omitted this in the results/discussion part when we re-mention reagents/kits.

We have added a paragraph of the possible result using other typer types in the Discussion (sentence no. 572)

We have also reviewed the language and made changes accordingly. 

We hope these changes meet the requirements by the reviewer and editor.

Reviewer 2 Report

a well written article, the introduction needs to be improved, especially with the role in immunotherapy, especially in lung cancer

Author Response

We would like to thank the reviewer and express our appreciation for the feedback.

Below is our reponse to the reviewer's comment:

Studies indicate that TMB may be associated with response to immunotherapy in patients with lung cancer (1,2).

Our study focuses on the technical validation of OCAP and TSO500, including TMB scoring. We have established a lung cancer biobank and, therefore, had access to these SCLC samples for the technical validation. None of these patients were enrolled in a clinical trial on immunotherapy or other personalised treatment. We have added a sentence in the method section (sentence no. 84) to state that our results did not impact the treatment of the patients. 

We have also slightly edited the sentence about TMB and immunotherapy in the introduction to be more modest. We are aware that TMB cannot be used as a solo biomarker for immunotherapy, especially in lung cancer. Since technical validation was our aim, the clinical utility of TMB or other NGS results in lung cancer was not described in more detail.

The results show that the two panels classified most samples (84%) with the same TMB category, and both panels could be applied in clinical trials on immunotherapy where TMB is required.

We hope these changes meet the requirements by the reviewer and editor.

References:

  1. Hellmann, M.D.; Ciuleanu, T.-E.; Pluzanski, A.; Lee, J.S.; Otterson, G.A.; Audigier-Valette, C.; Minenza, E.; Linardou, H.; Burgers, S.; Salman, P.; et al. Nivolumab plus Ipilimumab in Lung Cancer with a High Tumor Mutational Burden. N Engl J Med 2018, 378, 2093–2104, doi:10.1056/NEJMoa1801946.

  1. Hellmann, M.D.; Callahan, M.K.; Awad, M.M.; Calvo, E.; Ascierto, P.A.; Atmaca, A.; Rizvi, N.A.; Hirsch, F.R.; Selvaggi, G.; Szustakowski, J.D.; et al. Tumor Mutational Burden and Efficacy of Nivolumab Monotherapy and in Combination with Ipilimumab in Small-Cell Lung Cancer. Cancer Cell 2018, 33, 853-861.e4, doi:10.1016/j.ccell.2018.04.001.